# OpenReview forum: "Tulu 3: Pushing Frontiers in Open Language Model Post-Training"
_colmweb.org/COLM/2025/Conference — COLM 2025_

### Official Review · Reviewer_WQtE · 2025-05-08

**Rating:** 7
**Confidence:** 3
**Ethics Flag:** 1

**Summary:**

This work presents TULU3, a fully open-source post-training framework. The main contributions of this framework are as follows:

1. **Complete Open-Source**: The entire post-training process, including data cleaning, evaluation procedures, and training workflows, is fully open-sourced, which promotes transparency across all stages of the post-training process.

2. **Novel Post-Training Algorithm Pipeline**:
   - **SFT**: Initialized via a carefully designed mixed dataset for Cold Start.
   - **DPO**: Utilizes Direct Preference Optimization (DPO) to align the model with human preferences.
   - **RLVR**: Employs Reinforcement Learning with Verifiable Rewards (RLVR) to further improve performance on verifiable tasks such as code generation, mathematical reasoning, and instruction adherence.

3. **Superior Performance**: Through this fully reproducible post-training framework, the model demonstrates enhanced capabilities at every stage. Notably, the 405B model achieves performance comparable to closed-source models in certain benchmarks.

**Questions To Authors:**

1. Training reasoning capabilities often involves additional processes (e.g., DeepSeek-R1-Zero). How do the authors plan to incorporate reasoning-focused training into the post-training phase?

2. After RLVR fine-tuning, is there observable degradation in preference alignment? Have the authors explored the impact of swapping the DPO and RL training stages on final model performance?

**Reasons To Accept:**

1. This work enhances the transparency of the post-training phase, including various data-cleaning tricks and observations (e.g., n-gram methods proving more effective), the design of an unseen benchmark with leaked data removed to ensure model generalization, and a carefully structured post-training pipeline to improve diverse model capabilities. These contributions significantly advance the field's understanding of post-training paradigms.

2. The entire post-training process is reproducible and has achieved superior performance across models of varying sizes, providing highly credible reference value for post-training research.

**Reasons To Reject:**

1. The current approach of first aligning preferences via DPO training and then enhancing math and code capabilities through RLVR may present issues. For instance, RL training could potentially degrade preference alignment. This concern is further highlighted by the fact that the LLaMA4 report[1] adopted the opposite sequence—RL first, followed by DPO, since they consider DPO may limit further exploration.

2. The framework lacks a dedicated post-training design for reasoning models, including specific training phases and evaluation metrics (e.g., AIME). Given the widespread adoption of reasoning model training methodologies, this omission represents a notable gap.

[1] https://ai.meta.com/blog/llama-4-multimodal-intelligence/

---

> ### Author Response · Authors · 2025-05-31
>
> Thank you for the detailed review and for highlighting our novel, open and well-performing post-training recipe!
>
> - Regarding the sequence of preference alignment and RLVR: It is first important to note that Llama 4 was released after the submission of this paper (March 28th vs. April 5th). At the time of this submission, our initial investigation showed a slight improvement of running RLVR after the DPO step. Generally, we didn't see degradation on tasks after using RLVR.  However, we agree with the reviewer that the future work could show a detailed experimentation on the order of these steps.
> - Regarding Reasoning Models: the main purpose of this work is to establish post-training recipes for generalist language models (such as gpt4o or deepseek v3). Since the submission of this work, several lines of work study the role of RLVR to build reasoning models. We agree that future work should include developing post-training recipes for general-purpose reasoning models.

---

### Official Review · Reviewer_yFmp · 2025-05-10

**Rating:** 9
**Confidence:** 5
**Ethics Flag:** 1

**Summary:**

This paper describes the Tulu 3 LLM post-training recipe for English that involves several training steps with different types of data. The recipe and datasets include a) instructions with prompts and responses for supervised finetuning (SFT) b) instructions with prompts and preferences for Direct Preference Optimization (DPO) and c) a novel method called Reinforcement Learning with Verifiable Rewards (RLVR). In addition, Tulu 3 post-training recipe stablishes a multi-task evaluation benchmark for development and unseen evaluations.

**Questions To Authors:**

The final selection of the prompt dataset (Table 8) looks quite arbitrary. How did you select the final number of instructions for each dataset to be included in the mix? For instance, why just 50K instructions for SFT from OpenMathInstruct? Why 25.356 for DPO?

The post-training datasets included in Tulu 3 have very distinct licensing regimes:
https://drive.google.com/file/d/1HnIRFvRZeyCoVGwnKlrXx0cfDZSvnNES/view

In fact, most of them have been derived from proprietary LLMs (mostly OpenAI but also Meta) with quite restrictive licenses. For instance, several datasets derive from PersonaHub with CC-BY-NC-SA license and part of the dataset was created using Llama-3. Its license restricts using any part of the Llama models, including the response outputs, to train another AI model. No Robots license is also CC-BY-NC, etc. How this affects the Tulu 3 models released?

Please, check all citations to the appendixes. For instance, not all of them are referenced in the paper. Also remove the comment appearing in the caption of Table 15.

**Reasons To Accept:**

Language model post-training is possibly the most important step for developing advanced LLMs. Unfortunately, available open recipes for post-training lag behind proprietary ones which also remain undisclosed. Tulu 3 bridges the gap between open and closed post-training methods. The paper provides a complete recipe of four stages to obtain a  state-of-the art post-trained LLM. The authors also release final models trained on Llama 3.1 base versions, training and evaluation data, and code. In addition, the paper also introduces a novel last stage of the post-training recipe called Reinforcement Learning with Verifiable Rewards (RLVR) for training LLMs on tasks with verifiable answers such as mathematical problem-solving.

**Reasons To Reject:**

The paper derives from a very long technical report including many experiments, studies and results that now are placed in a very long and dense list of appendixes. Thus, the format of the resulting paper is quite difficult to follow.

---

> ### Author Response · Authors · 2025-05-31
>
> We thank the reviewer for highlighting that our paper bridges the gap between open and closed post-training methods, by releasing open recipes, code, data and an evaluation benchmark.
> As the reviewer pointed out, this paper is distilled from a much longer technical report. We will make sure to improve readability and to clean up the Appendix (thank you for pointing out the appendix - paper reference mismatches!), for the camera-ready version.
>
>
> Answers to the questions:
>
> On prompt selection and the number of prompts: The final number of prompts and their composition were determined according to the data mixing results, for both SFT and DPO.
> - For SFT, we ran careful data ablations to balance performance across our evaluations, and we found that adding “too much” data from one domain, for example math, degraded non-math evaluations, while not providing further benefit on math-specific evaluations. We therefore downsampled the prompts to maintain performance with fewer samples.
> - Similar concerns were valid for DPO. We additionally used smaller prompt subsets for DPO because of cost considerations and also because earlier experiments showed that scale matters less compared to the diversity of the responses (and some realities of needing to ship the model at some point even if we have more opportunities!).
>
> On dataset licensing: Licensing is a very important topic – thank you for further feedback on the area that is too often ignored! When sourcing prompts, we took consideration of the licenses of the original datasets, and we even manually tracked the provenance of the subsets of datasets and removed those with issues (for example ShareGPT). Regarding some of the specific datasets mentioned in the review:
> - For PersonaHub (CC-BY-NC-SA) we generated (prompt, response) pairs ourselves, conditioned on the personas and we discarded all persona description. Our data does not contain the personas that are covered by that license. For future models we’re exploring regenerating the persona’s to get out of this licensing grey area.
> - For the data derived from Llama: we apply our recipe to llama models and therefore decided that using llama derived data would be fine for our purposes. We comply with the llama3.1 license and named our models and data with the prefix “Llama-3.1”.
> - For data composed of other frontier model outputs: we follow standard practice of the community that has used and released many datasets with outputs from these models. In doing so, we try to remind users in the dataset cards that the data is subject to the terms and conditions of those providers, but historically there has been no meaningful enforcement of them.
> - For the No Robots dataset: We are removing that from future versions and will be releasing models without it ASAP.

---

> > ### Comment · Reviewer_yFmp · 2025-06-06
> >
> > Nice and useful contribution.
> >
> > However, the Tulu 3 post-training recipe shows how difficult it is to instruct a base model without relying on knowledge distillation from frontier models.

---

> > > ### Author Response · Authors · 2025-06-09
> > >
> > > This is true! While we could train an instruct model using only human-written data, this is significantly more costly than using synthetic data due to the relative lack of openly available high-quality human data already available for use. We also note that distillation from (often larger) models is now commonplace within industry even if human annotation was in budget. For example, Llama 4 [2], Qwen 3 [3], DeepSeek R1 [4] all made use of distillation to improve their smaller models. As such, it is clear that synthetic / distillation data is a key part of current post-training recipes. Our final recipe consists of a mix of human and distilled data, and we encourage researchers and especially companies to release more openly available human instruction following and preference data.
> > >
> > > [2] https://ai.meta.com/blog/llama-4-multimodal-intelligence/
> > > [3] Yang, A., Li, A., Yang, B., Zhang, B., Hui, B., Zheng, B., Yu, B., Gao, C., Huang, C., Lv, C. and Zheng, C., 2025. Qwen3 technical report. arXiv preprint arXiv:2505.09388.
> > > [4] Guo, Daya, et al. "Deepseek-r1: Incentivizing reasoning capability in llms via reinforcement learning." arXiv preprint arXiv:2501.12948 (2025).

---

### Official Review · Reviewer_BY8Z · 2025-05-17

**Rating:** 8
**Confidence:** 4
**Ethics Flag:** 1

**Summary:**

The paper presents a technically rigorous approach to post-training in general. Empirical results are comprehensive, covering various model sizes (8B, 70B, 405B) and a diverse set of benchmarks that goes beyond math reasoning, showing consistent improvement over baseline and existing open-source models. Experimental design is thorough and data curation is detailed. The main paper is clearly structured, where figures and tables effectively support the narrative.

I have a few questions regarding the introduced algorithm RLVR (Reinforcement Learning with Verifiable Rewards) and am open to discussion if I missed anything from reading:
1. What is the technical novelty of RLVR as compared to the previous iterative SFT approaches based on answer verification? For example, ReST-EM [1] collects self-training data by filtering model-generated responses using a binary reward (0/1) based on final answer correctness and then use for training. Essentially, such approach can be applied to data batches to achieve online RL. There has been a several other work similarly leveraging verifiable rewards iteratively [2-3].

2. What is the advantage of RLVR as compared to GRPO and its variants? Can they achieve similar performances?

[1] Beyond Human Data: Scaling Self-Training for Problem-Solving with Language Models

[2] Iterative Reasoning Preference Optimization

[3] V-STaR: Training Verifiers for Self-Taught Reasoners

(Minor) There are several formatting issues in the appendix that may need fixing (e.g. table 23 & 24)

**Questions To Authors:**

Is the offline RL algorithm DPO still necessary when online/on-policy RL algorithms (RLVR) are already employed? Many current post-training scheme proceed from SFT to GRPO or directly start with GRPO. I agree that starting RL from a better basis results in better final performances, but this better basis can be achieved in many ways such as curating better quality SFT data or iterative approaches. What is the intuition here to include DPO as an intermediate stage?

**Reasons To Accept:**

- The work presents notable efforts toward LLM post-training strategies. The comprehensive release of models, datasets, training recipes, and evaluation frameworks significantly contributes to reproducibility and will serve as a valuable resource for further research development.
- The paper is clearly structured where figures and tables effectively support its narrative.

**Reasons To Reject:**

- The novelty of RLVR is unclear to me. There have been previous work on iterative SFT based on answer verification, as well as GRPO that also relies on verifiable reward.
- The necessity of the DPO stage in a post-training scheme also needs more discussion. (See in question)

---

> ### Author Response · Authors · 2025-05-31
>
> We thank the reviewer for their thoughtful comments and we are happy to see that the reviewer appreciates the comprehensiveness and diversity of our experiments and results.
>
> - On the novelty of RLVR: Our approach to RLVR is inspired by prior work using RL for improving reasoning skills (e.g., Kazemnejad et al., 2024), and we further extend it to cover more domains, are the first to use asynchronous PPO for RLVR training, carefully ablate key aspects, and evaluate RLVR’s effect on a broad range of evaluations, not just limited to mathematics. More specifically,
>   - We show that RLVR itself can extend beyond comparing to ground-truth answers with our IFEval results, where there are no ground truth labels involved (as we instead provide reward based on how well the model’s output adheres to the given constraints).
>   - While prior approaches did indeed use iterative SFT, we instead use an efficient asynchronous PPO setup (section 6.2), and we cover more domains and evaluations than prior work.
>   - We ablate core aspects of the setup, including value model initialization, RM scores, and the effect of differing policy initializations.
>
>  We note that while there is indeed a current boom of ‘reasoning’ models, our work is prior to much of this work and also has a unique focus: rather than explicitly trying to target reasoning skills and reasoning skills alone, we instead focus on how to improve general non-thinking performance whilst maintaining (or improving) in domains beyond math. As such, we believe that our approach is novel (as said by reviewer yFmp) and contains many useful findings for the field.
> - On DPO: Our experiments (shown in tables 6 and 7) show that  the pipeline that includes the DPO step significantly improves a model’s general chat capabilities over the SFT step (+20 points on AlpacaEval 2, for example, for the 8B model). Additionally, we applied RLVR to a subset of domains that were verifiable, i.e. precise IF and math, and the DPO step was therefore needed to further strengthen the rest of the skills. Our experimentation showed that using RLVR after DPO is superior to using DPO after RLVR. For future work we could explore getting rid of the DPO step and instead only use RL (RLHF + RLVR) and we could experiment with the order of the pipeline too!
> The phrasing specifically of “Many current post-training scheme proceed from SFT to GRPO or directly start with GRPO.” is for reasoning models such as DeepSeek R1 or OpenAI’s o3. The training recipe introduced in this work and resulting models are designed for generalist models such as DeepSeek V3 V3 or GPT 4o. We plan to explore that style of training in future work.

---

> > ### Comment · Reviewer_BY8Z · 2025-06-11
> >
> > Thank the authors for the detailed response regarding RLVR and DPO! I appreciate the clearness and helpfulness of this paper in understanding the post-training schemes and look forward to the future explorations. I will increase my score from 7 to 8.

---

### Comment · Area_Chair_zeEn · 2025-06-04
**Discussion period**

Dear reviewers,

This is a reminder that the discussion period is currently in progress and will end on June 10th.

I encourage you to read the other reviews as well as the responses by the authors and engage in a discussion.

Thanks a lot!

- AC

---

### Decision · Program_Chairs · 2025-07-08

**Decision:**

Accept

**Comment:**

This paper introduces Tülu 3, a family of language models that were post-trained via SFT, DPO, and RL. The main focus of the paper is on the post-training recipe and all the details that go into (post)training language models to be competitive with state-of-the-art models.

All reviewers agree that the paper makes a significant contribution and in particular highlighted the extensive experimental results, ablation studies, and release of model weights, data, and code.

I agree with the reviewers and consider this paper a strong contribution for making the often times opaque language model development process more transparent.